# Extracellular Vesicles as Drug Carriers for Enzyme Replacement Therapy to Treat CLN2 Batten Disease: Optimization of Drug Administration Routes

**DOI:** 10.3390/cells9051273

**Published:** 2020-05-20

**Authors:** Matthew J. Haney, Yuling Zhao, Yeon S. Jin, Elena V. Batrakova

**Affiliations:** 1Center for Nanotechnology in Drug Delivery, University of North Carolina at Chapel Hill, Chapel Hill, NC 27599, USA; mjhaney@email.unc.edu (M.J.H.); yulingz@email.unc.edu (Y.Z.); 2Eshelman School of Pharmacy, University of North Carolina at Chapel Hill, Chapel Hill, NC 27599, USA; ysjin@email.unc.edu

**Keywords:** Batten disease, brain bioavailability, drug delivery, extracellular vesicles, neuroinflammation

## Abstract

CLN2 Batten disease (BD) is one of a broad class of lysosomal storage disorders that is characterized by the deficiency of lysosomal enzyme, TPP1, resulting in a build-up of toxic intracellular storage material in all organs and subsequent damage. A major challenge for BD therapeutics is delivery of enzymatically active TPP1 to the brain to attenuate progressive loss of neurological functions. To accomplish this daunting task, we propose the harnessing of naturally occurring nanoparticles, extracellular vesicles (EVs). Herein, we incorporated TPP1 into EVs released by immune cells, macrophages, and examined biodistribution and therapeutic efficacy of EV-TPP1 in BD mouse model, using various routes of administration. Administration through intrathecal and intranasal routes resulted in high TPP1 accumulation in the brain, decreased neurodegeneration and neuroinflammation, and reduced aggregation of lysosomal storage material in BD mouse model, CLN2 knock-out mice. Parenteral intravenous and intraperitoneal administrations led to TPP1 delivery to peripheral organs: liver, kidney, spleen, and lungs. A combination of intrathecal and intraperitoneal EV-TPP1 injections significantly prolonged lifespan in BD mice. Overall, the optimization of treatment strategies is crucial for successful applications of EVs-based therapeutics for BD.

## 1. Introduction

Lysosomal storage disorders (LSDs) are severe neurodegenerative diseases that are characterized by deficiency in a specific lysosomal enzyme, leading to progressive neurodegeneration. One of this group, CLN2 Batten disease (BD), results from mutations in TPP1 gene, causing an insufficiency or complete lack of a soluble lysosomal enzyme, tripeptidyl peptidase-1 (TPP1) [1]. Without functional TPP1, neurons develop inclusions of storage material; the retina and central nervous system (CNS) undergo progressive degeneration [2], triggering the loss of neurological functions and vision [3]. Furthermore, the accumulation of storage material throughout the body leads to peripheral organs damage [4]. Children with BD typically begin experiencing seizures between the ages of two and four years old, preceded in the majority of cases by language development delay [5]. The disease progresses rapidly, with most affected children losing the ability to walk and talk by approximately 6 years of age. Eventually, children with BD become blind, bedridden, and lose all cognitive functions. Thus, a successful enzyme replacement therapy (ERT) that provides the delivery of functional TPP1 to the CNS and other organs is of great importance.

A major drawback in the treatment of LSDs has been encountered due to the restricted transport of a therapeutic enzyme to the CNS. The existence of multiple biological barriers that prohibit effective drug delivery to neurons caused low efficacy of ERT. Recently, the first FDA-approved treatment, intraventricular infusions of a recombinant form of human TPP1, Brineura, was shown to provide symptomatic improvements in pediatric patients [5,6,7]. The administration of Brineura in patients with CLN2 disease has led to significant reductions in the rate of decline of motor and language functions in comparison with a natural history population. Unfortunately, this invasive procedure carries a high risk of adverse effects. In addition, long biweekly infusions of TPP1 have a low patient adherence to treatment that is a major problem in health care, especially for kids [8]. The less invasive parenteral administration provides no therapeutic benefits, since biological barriers in the body, and specifically, the blood brain barrier (BBB), severely restrict transport of macromolecules —including TPP1—to the brain. In addition, there is a significant risk of immune response to the repeated TPP1 injections.

To circumvent this problem, we developed a novel, biomimetic delivery system using macrophage-derived extracellular vesicles (EVs), that are capable of TPP1 transport to the brain. The field of drug delivery has benefited from a recent boom in academic research surrounded EVs. These naturally occurring nanosized vesicles are composed of two types, exosomes and microvesicles. The unique properties of EVs can be attributed to the biogenesis; exosomes are initially produced by invagination of the endosomal membrane to create multivesicular bodies (MVB), and microvesicles bud directly from the plasma membrane. Consequently, exosomes and microvesicles have endosomal and plasma membrane origin, respectively. EVs can provide unprecedented opportunities for the transport of therapeutic proteins to target cells. They have a low risk of long-term immunogenicity and cytotoxicity, which is a disadvantage for synthetic nanocarriers. Of note, they have a high capability to cross biological barriers, especially under inflammatory conditions [9].

We previously demonstrated that macrophage-derived EVs home to regions of inflammation and neurodegeneration accumulate at therapeutically relevant amounts, and deliver therapeutic proteins to the CNS [10,11,12]. We have developed different methods for the isolation, purification, and characterization of EVs [10,11,12,13,14,15], as well as various techniques for loading therapeutic proteins into EVs, including the transfection of parental EV-producing macrophages with encoding therapeutic protein plasmid DNA (*p*DNA), and drug loading into naive drug-free EVs [16]. As a result, EVs-based formulations of TPP1 (EV-TPP1) with a high loading capacity, sustained release, and efficient preservation against degradation and clearance were developed and manufactured [16]. 

Herein, we investigated biodistribution and therapeutic efficacy of EV-TPP1 obtained by incorporation of the enzyme into naïve EVs under sonication, and administered through various routes in BD mouse model, CLN2 knock-out (CLN2 KO) mice. The specific advances to the field contributed by this body of work can be outlined in three main points. First, intrathecal (*i.t.*) route of administration provided the highest CNS levels of EV-TPP1 in CLN2 KO mice, reduced deposition of lysosomal storage material in brain tissues resulting in decreased neurodegeneration and neuroinflammation. Second, treatment with EV-TPP1 via intraperitoneal (*i.p.*) and intravenous (*i.v.*) routes ensued high accumulation of the enzyme in peripheral organs: liver, kidney, spleen, and lungs. Finally, combination of *i.t.* and *i.p.* injections of EV-TPP1 significantly increased lifespan in CLN2 KO mice. In summary, using EVs nanocarriers is a promising strategy for successful TPP1 delivery to target cells, with a potential to revolutionize the treatment for BD.

## 2. Materials and Methods

### 2.1. Reagents 

Recombinant full-length Human TPP1 protein (lot #BIQE03, UniProtKB—O14773 (TPP1_HUMAN)) was a generous gift from BioMarin Pharmaceutical Inc. (Novato, CA, USA). Lipophilic fluorescent dyes, 1,1′-Dioctadecyl-3,3,3′,3′-tetramethylindo-carbocyanine iodide (DIR), and 1,1′-Dioctadecyl-3,3,3′,3′-tetramethylindo-dicarbocyanine (DID), and Alexa Fluor 555 labeling kit were purchased from Thermo Fisher Scientific (Waltham, MA, USA). A nuclear dye, 4′,6-diamidino-2-phenylindole dihydrochloride (DAPI), and fluorescent substrate for TPP1, ala-ala-phenilalanin-7-amido-4-methylcoumarin (AAF-AMC) were obtained from Sigma-Aldrich (St. Louis, MO, USA). Alexa Fluor-555 protein labeling kit was purchased from Invitrogen (Carlsbad, CA, USA). FITC-conjugated mouse antibodies to LAMP1 were purchased from BD Biosciences (San Diego, CA, USA). Primary Anti-NeuN antibody (ab77487, neuronal marker), rabbit polyclonal anti-GFAP antibody (ab7260, astrocytosis marker), and anti-ATP synthase C antibody (ab181243, lysosomal storage body marker) were purchased from Abcam (Cambridge, MA, USA). Secondary antibody, goat anti-rabbit IgG H+L Alexa Flour 488 (A-11008), were purchased from Invitrogen (Carlsbad, CA, USA). Murine macrophage colony-stimulating factor (MCSF) was purchased from Peprotech Inc. (Rocky Hill, NJ, USA). Cell culture medium and fetal bovine serum (FBS) were purchased from Gibco Life Technologies (Grand Island, NY, USA).

### 2.2. Cells and Animals 

Primary bone marrow-derived cells extracted from murine femurs of C57BL/6 mice, as described in [17], were cultured for 10 days in the media supplemented with 1000 U/mL macrophage colony-stimulating factor (MCSF), to obtain primary bone marrow-derived macrophages (BMM). The purity of monocyte culture was determined by flow cytometry using FACS Calibur (BD Biosciences, San Jose, CA). IC21 cell line derived by the transformation of normal C57BL/6 mouse peritoneal macrophages with SV40 (cat # TIB-186) was purchased from American Type Culture Collection (ATCC, Manassas, VA, USA), and cultured in bioreactors in Dulbecco’s modified Eagle’s medium (DMEM) (Hyclone, South Logan, UT, USA), supplemented with 10% FBS, and 1% (*v*/*v*) of both penicillin and streptomycin. Mouse primary cultured cortical neurons were isolated from mouse pups cortex and hippocampus as described [18].

Late-infantile neuronal ceroid lipofuscinosis (LICL) mice with mutations of the CLN2 gene encoding a soluble lysosomal enzyme TPP1 were used as the in vivo model of BD [19]. The TETRA-ARMS design (Appendix A) described earlier [16] was used for CLN2 genotyping. Specifically, mutant: 266 bp; wild-type: 493 bp; and locus: 704 bp bands were visualized to identify mutant KO and wild-type mice, with inner primers that bind to either the wild-type (WT) or mutant (KO) sequence (Appendix A). It was reported that the first manifestations of storage pathology can be visible on light microscopic levels as early as at 35–40 days of age, although major clusters of inclusions-lysosomes are developed by 90–100 days of age [19]. Mutant KO and wild type (WT) mice were housed in a temperature and humidity-controlled facility on a 12 h light/dark cycle, and food and water were provided ad libitum. Mice were treated in accordance to the Principles of Animal Care outlined by National Institutes of Health and approved by the Institutional Animal Care and Use Committee of the University of North Carolina at Chapel Hill.

### 2.3. EVs Isolation and Characterization 

EVs were isolated from BMM media by ultracentrifugation, as described earlier [20]. Briefly, EVs were harvested from the conditioned media of macrophages seeded into Bioreactor (3.6 × 10^8^ cells/flask) and cultured in EV-depleted media for 2 days, using gradient centrifugation and described in [10]. The culture supernatants were cleared of cell debris and large vesicles by sequential centrifugation at 300× *g* for 10 min, 1000× *g* for 20 min, and 10,000× *g* for 30 min, followed by filtration using 0.2 μm syringe filters. Then, the cleared sample was spun at 100,000× *g* for one hour to pellet the EVs, and supernatant was collected. The collected EVs (10^11^–10^12^ EVs/flask) were washed twice with phosphate buffer solution (PBS). To avoid contamination by the FBS-derived EVs, FBS was spun at 100,000× *g* for 2 h, to remove EVs before the experiment. The particle concentration and size were determined by nanoparticle tracking analysis (NTA) using NanoSight 500 Version 2.2 (Wiltshire, UK). For the size measurements, EVs were dispersed at concentration ~3 × 10^10^ particles/mL in Phosphate-Buffered Saline (PBS). Protein concentrations were determined using BCA kit (Pierce Biotechnology, Rockford, IL, USA). The levels of proteins constitutively expressed in EVs (CD63, TSG101, and HSP90) were identified in EV-TPP1 formulations by Western blot analysis, using Wes^™^ (ProteinSimple, San Jose, CA, USA). The protein bands were detected with CD63 primary monoclonal antibodies (Novus Biologicals, Centennial, CO, USA; 1:1000 dilution, #NPB2-67425), or TSG101 monoclonal antibodies (Novus Biologicals, #NPB2-67884), or HSP90 monoclonal antibodies (Novus Biologicals, #NPB2-67395), and secondary HRP-conjugated rabbit anti-goat IgG-HRP (Santa Cruse, CA, USA; 1:5000 dilution). The purity of EVs preparations was estimated by a ratio of nano-vesicle counts to protein concentration [21]. The calculated ratio of 2.0 × 10^10^ particles/µg protein indicated near 0% protein contamination.

### 2.4. Accumulation of EV-TPP1 in Primary Neurons by Confocal Microscopy

TPP1 was labeled with Alexa Fluor 555 Protein Labeling Kit (Thermo Fisher Scientific), according to the manufacturers protocol. Briefly, TPP1 was dissolved in 0.1 M sodium carbonate buffer, pH 8.5 (1 mg/mL) and added to Alexa Fluor 555 solution for one hour at RT. Labeled TPP1 was purified from low molecular weight residuals by gel filtration on a Sephadex G-25 column (1 × 20 cm) in PBS at elution rate 0.5 mL min^−1^, and lyophilized. 

EVs were loaded with fluorescently-labeled Alexa 555-TPP1 using the sonication procedure [16]. Briefly, 0.5 mL EVs suspension (10^11^ particles/mL) were supplemented with 5 µL Alexa 555-TPP1 solution (final concentration: 1 µg/mL), sonicated on Fisher Scientific FB 505 sonicator with 20% amplitude, and then kept at room temperature (RT) for 1 h, to allow membrane recovery. According to transmission electron microscopy (TEM) studies, the integrity of EVs was preserved after sonication, as reported earlier [10,16]. EV-Alexa 555-TPP1 formulation was purified from non-incorporated enzyme using size-exclusion chromatography. Primary neurons were incubated with Alexa 555-TPP1 incorporated into EVs (10^11^ particles/mL; 0.2 mg/mL Alexa 555-TPP1), or Alexa 555-TPP1 alone (0.2 mg/mL), or TPP1-free media for 4h at 37 °C. Following incubation, the cells were washed with 2× PBS, fixed, permeabilized, and incubated with FITC-LAMP1 antibodies. The staining solution was removed, cells were washed 2× PBS, and stained for nuclei with DAPI prior to the imaging [11]. Labeled cells were examined by a confocal fluorescence microscopic system ACAS-570 (Meridian Instruments, Okimos, MI, USA), with argon ion laser and corresponding filter set. Digital images were obtained using the CCD camera (Photometrics) and Adobe Photoshop software. Quantification of immunostaining was performed with Image software, utilizing JACoP plugins to calculate Pearson’s co-localization coefficients [22]. A comparison was performed on 10–20 sets of images acquired with the same optical settings. 

### 2.5. EVs Biodistribution Studies in BD Mouse Model by Bioimaging and Infrared Spectroscopy (IVIS) 

To track EVs nanocarriers in BD mice, EVs released by BMM were labeled with DIR, a lipophilic near-infrared fluorescent cyanine dye (emission peak of 790), according to manufacturer’s protocol. The fluorescence spectrum of this dye allows efficient penetration through the bones and tissues; therefore, DIR is the most appropriate for the imaging in living animals. Briefly, DIR stock solution in ethanol was added to EVs suspension (final concentration 2 µM) and incubated for 20 min at 37 °C Then, DIR-EVS were span down at 100,000× *g* to separate from non-incorporated dye, and further purified on Nap 10 column. To reduce fluorescence quenching by fur and autofluorescence from solid diet, CLN2 KO mice (1 month old) were shaved and kept on a liquid diet for 48 h prior to the imaging studies. Then, BD mice (*N* = 4) were injected with DIR-EVs through: *i.v.* (6 × 10^11^ particles/200 µL/mouse), or *i.p.* (6 × 10^11^ particles/200 µL/mouse), or *i.t.* (1.5 × 10^11^ particles/50 µL/mouse), or *i.n.* (6 × 10^10^ particles/20 µL/mouse). For the *i.v.* administration route, mice were injected with the EV-based formulation intra-tail vein, with the use of a restraining device. For the *i.p.* administration route, mice were injected into the left lower quadrant of the abdomen without a restraining device. For the *i.t.* administration route, mice were anesthetized with 3% isoflurane, and the fur at the posterior end was shaved. The EV-based formulation was slowly injected between the groove of L5 and L6 vertebrate. For *i.n.* administration route, mice were anesthetized with 3% isoflurane, and EV-based formulation was slowly dispensed into each nostril of the mouse, using a micropipette. Animals were imaged by Xenogen IVIS Optical Imaging System up to 480 h. At the end point, animals were sacrificed and perfused as described earlier [23], main organs were removed, washed, post-fixed in 10% phosphate-buffered paraformaldehyde, and evaluated by IVIS Aura software (Spectral Instrument Imaging, Tucson, AZ, USA). 

### 2.6. TPP1 Accumulation Studies in BD Mouse Model by Confocal Microscopy

CLN2 KO mice (1 month old, *N* = 4) were injected with fluorescently labeled Alexa 555-TPP1 loaded by sonication into non-labeled EVs, or DID-labeled EVs with incorporated non-labeled TPP1 through *i.v*., *i.p*., *i.t*., or *i.n*. routes in the doses used for IVIS studies. DID is the far-red fluorescent dye that can be detected in tissues by confocal microscopy without high background green fluorescence. CLN2 KO mice injected with TPP1 alone were used as controls. Then, 72 h later, mice were sacrificed, perfused with 4% PFA according to standard protocol, and the main organs (brain, liver, kidney, spleen, lungs, and spinal cord) were harvested and fixed in 4% paraformaldehyde overnight. Then, tissues were transferred in 30% sucrose solution for 24 h to enable cryoprotection of tissues and embedded in Tissue-Tek^®^ OCT. The 10-micron cryosections of each organ were mounted on slides, washed 3× PBS/Tween 5min/wash ddH_2_O, and covered using Vectashield Hardset mounting media with DAPI. The accumulation of fluorescently labeled TPP1 or EVs were visualized by a confocal fluorescence microscopic system ACAS-570 and corresponding filter set. All images were taken from the exact same brain region (parietal lobe), with the controlled laser exposure and image capture times. The quantification of immunostaining was performed with Image software, utilizing JACoP plugins. A comparison was performed on 20–30 sets of images, acquired with the same optical settings. 

### 2.7. Immunohistochemical Analyses

CLN2 KO mice (1 month old, *N* = 6) were treated with EV-TPP1 through different routes twice a week, using: *i.v.* and *i.p.* injections (4.5 × 10^10^ particles/150 µL/mouse; 15 mg/kg TPP1), or *i.t.* injections (1.5 × 10^10^ particles/50 µL/mouse; 5 mg/kg TPP1, or *i.n.* administration (0.6 × 10^10^ particles/20 µL/mouse; 2 mg/kg TPP1). BD mice and WT littermates *i.v.* injected with saline were used in control groups. Three weeks later, mice were sacrificed, perfused, brain was harvested, and slides were stained with Ab to neurons (NeuN, ab177487, 1:200 dilution), for the assessment of the neuroprotection effect, or primary antibody anti-GFAP (ab7260, 1:500 dilution), to activate astrocytes for levels of neuro-inflammation; or primary antibody to subunit c of mitochondrial ATP synthase (ab181243, 1:500 dilution), for the accumulation of lysosomal storage bodies. Then, brain slides were stained with secondary antibody goat anti-rabbit IgG H+L Alexa Flour 488 (ab11008) for 1h at RT in the dark. Immunohistochemical analysis was performed in 10 μm thick brain sections [24]. All images were taken from the exact same brain region (parietal lobe), with the controlled laser exposure and image capture times. It was reported that this part of the brain undergoes degeneration at earlier stages of the disease [25]. Quantification of the fluorescence levels was performed as the function of the positive area by ImageJ software (Java 1.8.0_112, free access provided by National Institute of Health). A comparison was performed on 30–40 sets of images acquired with the same optical settings. 

### 2.8. Survival in LINCL Mice Studies

Non-labeled TPP1 was incorporated into EVs using the sonication procedure described above and enzymatic TPP1 activity was measured by spectrofluorimetry. Briefly, 20 μL of EVs or free TPP1 were first added to 60 µL of activation buffer (50 mM acetate, 100 mM NaCL, 0.1% Triton X100, 0.4 mg/mL saponin, pH 3.5) and incubated for 1 h at 37 °C. Then, 10 µL of these solutions were mixed with 40 μL of the assay buffer (50 mM acetate, 100 mM NaCL, 0.1% Triton X100, pH 5.0) and 50 μL of AF-AMC in the assay buffer (substrate final concentration 400 µM, EVs final concentration 4.5 × 10^11^ particles/mL, or free TPP1 final concentration 3 × 10^−9^ M). The reaction rate was determined by recoding AMC fluorescence (λex = 380 nm, λem = 460 nm), using Spectramax for 10 min and employing the AMC calibration curve. Of note, sonication loading procedure did not decrease TPP1 activity, as validated by spectrofluorimetry. 

CLN2 KO mice (one-week-old, *N* = 6) were injected through *i.p*. route with EV-TPP1 formulation loaded by sonication (4.5 × 10^10^ particles/150 µL/mouse; 15 mg/kg TPP1) twice a week, over six weeks. Another group was treated with a combination of *i.p*. injections (same schedule as described above), and *i.t*. injections of EV-TPP1 (1.5 × 10^10^ particles/50 µL/mouse; 5 mg/kg TPP1), for five weeks. BD mice were treated twice a week, over six weeks, with: saline (150 µL/mouse), or empty (sham) EVs (4.5 × 10^10^ particles/150 µL/mouse), or TPP1 alone (15 mg/kg TPP1) were used as controls. Finally, WT littermates treated with saline were used in another control group. The survival in each animal group was recorded for over three months.

### 2.9. Statistical Analysis

Results are reported as mean ± SEM of 3–5 independent experiments. Tests for significant differences between the groups were performed using a one-way ANOVA, followed by the post hoc test for multiple comparisons (Fisher’s pairwise comparisons), using GraphPad Prism 5.0 (GraphPad software, Inc., La Jolla, CA, USA). A minimum *p* value of 0.05 was chosen as the significance level for all tests.

## 3. Results

### 3.1. Manufacture and Characterization of EV-TPP1

EVs were collected from conditioned media of autologous bone marrow-derived macrophages (BMM), characterized according to MISEV2018 guidelines as described earlier [16], loaded with TPP1 by optimized sonication procedure, and purified from non-incorporated TPP1 by size-exclusion chromatography [16]. The obtained EV-TPP1 formulation was characterized for size and concentration by nanoparticle tracking analyses (NTA); EV-specific proteins–by WES™, and TPP1 enzymatic activity–by spectrofluorometric assay with fluorescent substrate, AAF-AMC. According to NTA, 6.5 × 10^12^ particles/mL were collected in total from one bioreactor, with average min size 115.0 ± 8.3 nm, and mode 96.7 ± 7.2 nm. The abundance of specificity to EVs markers (CD63, TSG101, and HSP90) was confirmed by Wes (Appendix A). The TPP1 enzymatic activity was estimated as 44,128 ± 2253 mU/min/10^11^ particles. 

### 3.2. EVs Promote TPP1 Accumulation in Primary Neurons In Vitro

TPP1 is enzymatically inactive; however, upon acidification in lysosomes, it is proteolytically processed and concomitantly acquires enzymatic activity. Therefore, the efficient delivery of TPP1 into lysosomes is crucial for the therapeutic efficiency. This transport can be facilitated in the format of EVs, natural nanoparticles. EVs display tetraspanins and integrins at their surface, enhancing attachment and accumulation in target cells [26,27,28]. Furthermore, they enter target cells, in part by endocytosis that followed by their transport to lysosomal compartments [16].

To assess the effect of EVs nanocarriers on TPP1 intracellular accumulation in target neural cells, primary neurons were isolated from P_0_ pups of CLN2 KO mice, and cultured for 10 days until maturation. The cells were exposed to fluorescently labeled Alexa 555-TPP1 loaded into EVs (EV-Alexa 555-TPP1; 10^11^ particles/mL; 0.2 mg/mL Alexa 555-TPP1), or TPP1-Alexa 555 without nanocarriers (0.2 mg/mL) for 4 h, and examined by confocal microscopy (Appendix A), respectively). Cells treated with TPP1-free media were used as a control (Appendix A). Confocal images suggested that EVs facilitated TPP1 accumulation in primary neurons. About 54% TPP1 incorporated into EVs were co-localized with lysosomes after the 4h incubation period.

### 3.3. Biodistribution of EVs nanocarriers in BD Mice by Bioluminescence Imaging (IVIS) 

We evaluated four different administration routes, intravenous (*i.v.*), intraperitoneal (*i.p.*), intrathecal (*i.t.*), and intranasal (*i.n*.) injections of EVs nanocarriers in BD mouse model (CLN2 KO mice, *N* = 4). To visualize EVs biodistribution, their lipid bilayers were labeled with a hydrophobic dye DIR. In this experiment, we used maximal dose and volume of EVs suspension for each route of administration (specifically, 6 × 10^11^ particles/200 µL/mouse for *i.v.* and *i.p.* injections, or 1.5 × 10^11^ particles/50 µL/mouse for *i.t.* injection, or 0.6 × 10^11^ particles/20 µL/mouse for *i.n.* injection). The DIR- EVs distribution was examined by bioluminescence in live animals using IVIS (Figure 1).

Significant levels of DIR-EVs in BD mouse brain were recorded at 24–120 h time frame after single *i.v.* (Figure 1a), and *i.p*. (Figure 1b) injections. Importantly, DIR-EVs were detected in the mouse brain up to 20 days after *i.v.* administration. Less, but still detectable, EVs fluorescence in the brain was recorded after *i.t.* administration (Figure 1c), and low if any fluorescence for *i.n.* injection (Figure 1d). Lesser fluorescent signals at the first six hours on all prone images might indicate that most DIR-EVs were circulated in the bloodstream and accumulated in main excretion organs, liver, spleen, and kidney, as seen on supine images on Appendix A.

The quantification of fluorescence levels of DIR-EVs signal in the brain area of BD mice was assessed on IVIS images using Aura software (Figure 2a). For all routes of administration, the maximal DIR-EVs signal in the brain was detected at 72 h after administration and decreased in the row: *i.v.* > *i.p.* > *i.t.* > *i.n*. injections (Figure 2a). One should keep in mind that the *i.t.* injected dose was four times, and *i.n.* injection was ten times lesser than those injected via *i.p.* and *i.v.* administration routes. Additionally, imaging in living animals does not allow one to discriminate whether or not fluorescence levels of DIR-EVs represent the amount of nanocarriers in the blood or the brain tissues. Therefore, this observation may result from both an accumulation of DIR-EVs in the brain tissues, as well as in the blood circulation.

To address this issue, mice were sacrificed at the endpoint of the experiment (480 h), perfused to eliminate the EVs-carriers in the blood, and main organs (i.e., liver, lungs, spleen, kidney, and brain) were imaged by IVIS (Figure 2b). The quantification of fluorescence levels on the images of postmortem organs assessed by Aura software indicated the high accumulation of DIR-EVs in liver, and kidney, when mice were treated through *i.v*. and *i.p.* routes (Figure 2c). Surprisingly, the highest amount of DIR-EVs in the brain was recorded after *i.t.* injection, although the administration dose was four times lesser than for *i.v.* and *i.p.* injections (Figure 2c, insert). This suggests that the *i.t.* administration appears to be the best for CNS drug delivery using EVs nanocarriers. We hypothesized that *i.t*. route allowed one to bypass EVs entrapment in peripheral organs, and therefore provided better brain bioavailability. Nevertheless, one should keep in mind that quantification IVIS images of isolated organs reflects total fluorescence of the whole organ, therefore high fluorescence count in liver, for example, may be not only due to the greater accumulation of DIR-EVs in this organ, but also to the overall larger liver mass compared to other organs. 

### 3.4. Effect of the Administration Route on Accumulation of TPP1 Loaded into EVs in BD Mouse Brain 

To identify the administration route that provides the most efficient brain delivery of TPP1 incorporated into EVs nanocarriers, fluorescently-labeled EV-TPP1 formulations were prepared by sonication procedure using DID-EVs (10^11^ particles/mL), and Alexa 555-TPP1 (0.2 mg/mL), and administered into *Cln2* KO mice (*N* = 6), via the four various routes described above. Specifically, mice were injected with fluorescently-labeled DID-EVs with incorporated non-labeled TPP1; or fluorescently labeled Alexa 555-TPP1 incorporated into non-labeled EVs, or Alexa 555-TPP1 alone (without EVs nanocarriers). Of note, similar to IVIS experiments, the maximal volume of EV-TPP1 suspension allowed for each route of administration was used (specifically, 200 µL for *i.v.* and *i.p.* injections, 50 µL for *i.t.* injection, and 20 µL for *i.n.* injections). At the optimal time point determined in IVIS studies (72 h), mice were sacrificed, perfused; brains were harvested, sliced, mounted on slides with nuclei staining DAPI, and examined by confocal microscopy (Figure 3). The representative images of brain slides of BD mice injected with DID-EV-TPP1, or EV-Alexa 555-TPP1 showed significant accumulation of both the EVs nanocarriers (Figure 3a) and incorporated TPP1 (Figure 3b) in the brain, after all examined administration routes. Importantly, the similar patterns of EVs and TPP1 distribution in case of EVs labeled through the incorporation of hydrophobic DID dye into EVs membranes (Figure 3a), or TPP1 labeled through covalent conjugation with Alexa 555 (Figure 3b) confirmed that recorded fluorescence reflects the distribution of the nanocarriers and TPP1, but not the leaked residual fluorescent dyes alone. Little, if any, fluorescent signal of Alexa 555-TPP1 was recorded in the case of TPP1 injection without nanocarriers (Figure 3c). In accordance with IVIS data, a single *i.t.* injection of EV-TPP1 provided the topmost levels of EVs nanocarriers (Figure 3a), as well as TPP1 incorporated into EVs (Figure 3b), in the brain of BD mice. The distribution of TPP1 delivered in EVs to the different brain regions through *i.t.* injection is presented on Appendix A. A significant amount of TPP1 was delivered by EVs nanocarriers to all examined brain regions, especially hypothalamus, thalamus, and pons. The quantification of fluorescent levels on confocal images confirmed that the *i.t.* route is the optimal one for the CNS delivery of TPP1 (Figure 3d).

### 3.5. Effect of the Administration Route on EV-TPP1 Therapeutic Efficacy in the BD Mouse Brain 

To assess neuroprotective effects in the BD mouse model, CLN2 KO mice (1 month old, *N* = 6) were treated with EV-TPP1 two times a week for three weeks with EV-TPP1 through *i.v.* or *i.p*. routes (4.5 × 10^10^ particles/150 µL/mouse; 15 mg/kg TPP1), or *i.t.* route (1.5 × 10^10^ particles/50 µL/mouse; 5 mg/kg TPP1), or *i.n.* route (0.6 × 10^10^ particles/20 µL/mouse; 2 mg/kg TPP1). BD mice and WT mice injected with saline were used as positive and negative controls, respectively. Saline-injected BD mice in the control group showed a prominent accumulation of cytoplasmic storage material within the lysosomal-endosomal compartments (Figure 4a), increased neuroinflammation (Figure 4b), and as a result, extensive neuronal degeneration (Figure 4c). In contrast, treatments with EV-TPP1 via all tested administration routes significantly diminished the accumulation of lysosomal storage bodies (Figure 4a), decreased neuroinflammation down to the levels of healthy WT animals (Figure 4b), and improved neuronal survival (Figure 4c). Importantly, the quantification of fluorescent levels of the confocal images for each route of administration indicated the superior therapeutic effects in CNS of EV-TPP1 formulations administered through *i.t.* and *i.n.* routes (Appendix A). Together, these data indicate that EVs can accomplish delivery of TPP1 to the brain in therapeutically sufficient amounts, and protect neurons in CLN2 KO mice, especially through *i.t.* and *i.n.* injections, even in spite of the lesser doses of EV-TPP1.

### 3.6. Effect of the Administration Route on Accumulation of TPP1 Loaded into EVs in Peripheral Organs of BD Mice 

Along with the brain tissues, the delivery of TPP1 to main peripheral organs is of great importance for successful ERT. We examined the accumulation of Alexa 555-TPP1 incorporated into EVs in liver, kidney, spleen, and lungs, as well as in the spinal cord in CLN2 KO mice that were treated via various administration routes described above (Figure 5). Mice were injected with the same doses of EV-Alexa 555-TPP1, and slides of the main organs were processed as described in Figure 3. Representative confocal images of main organs (Figure 5a–e), as well as the quantification of Alexa 555-TPP1 signals (Figure 5f), indicated that *i.v.* and *i.p.* administrations provided the largest amount of TPP1 in peripheral organs, liver, kidney, spleen and lung (Figure 5a–d,f). In addition, the administration of EV- TPP1 through the *i.t.* route resulted in high levels of the enzyme in the spinal cord of BD mice (Figure 5e,f).

### 3.7. Administration of EV-TPP1 through Combination of i.t. and i.p. Routes Prolonged Lifespan in BD mice

Based on the obtained results, we selected a combination of two routes that provided the high accumulation of TPP1 in the brain (*i.t.* injections) and peripheral organs (*i.p.* injections), and examined therapeutic effects of this ERT regimen on lifespan in BD mouse model (Figure 6). One group of CLN2 KO mice (one-week-old) was injected through *i.p.* route with EV-TPP1 formulation (4.5 × 10^10^ particles/150 µL/mouse; 15 mg/kg TPP1) twice a week, over six weeks. Another group was treated with a combination of *i.p.* injections (same schedule as described above), and *i.t.* injections of EV-TPP1 (1.5 × 10^10^ particles/50 µL/mouse; 5 mg/kg TPP1) once a week, starting from week two of age, for five consecutive weeks. Three control groups of BD mice treated with saline, or empty (sham) EVs (1.5 × 10^10^ particles/50 µL/mouse), or TPP1 alone (5 mg/kg) were *i.p.* injected twice a week, over six weeks. Another control group of WT mice was injected with saline. The survival was recorded for over three months (Figure 6). Treatments with EV-TPP1, especially via a combination of *i.p.* and *i.t.* routes, resulted in a significant increase in lifespan compared to control BD mice treated with saline, or with TPP1 alone. Treatment with sham EVs produced subtle, but not significant, increases in the life spans of BD mice.

## 4. Discussion

No specific treatment is known that can reverse symptoms of any form of BD, yet ERT allows one to alleviate the symptoms, and greatly modify or attenuate the progressive development of mental retardation, movement disorders, and behavioral changes [29]. The disease affects multiple organ systems, however the delivery of enzymatically active TPP1 to the CNS is a top priority. To accomplish CNS transport, we developed a drug delivery system using macrophage derived EVs as vehicles for TPP1. Bio-inspired, non-viral nanoparticles—EVs—can address the main limitations of current treatment regimens and would achieve both a robust accumulation of TPP1 in the brain, and a sustained drug delivery to the target neural cells. Furthermore, we demonstrated earlier that macrophage derived EVs can inherit some properties of their parental cells, and target regions of inflammation and neurodegeneration through interactions with ICAM-1 receptors overexpressed on the inflamed endothelium [10,11,12]. We evaluated different methods of TPP1 incorporation and identified one of the best approaches, i.e., sonication of EVs in the presence of TPP1 [16].

The present work was focused on the optimization of treatment regimens for EV-TPP1 in a BD mouse model, CLN2 KO mice. Importantly, the routes of administration can greatly affect the bioavailability of the therapeutic enzyme by changing the number of biological barriers the drug must cross, or by changing the exposure of the drug to pumping and metabolic mechanisms. Herein, we tested different routes to identify the best way for efficient enzyme delivery to the brain and main organs in BD mice. First, we demonstrated a prolonged substantial accumulation of EVs nanocarriers in the brain and main organs in living animals by IVIS. Four routes including *i.v., i.p., i.t.*, and *i.n.* administrations were investigated. The fluorescent signals of DIR-EVs were detected up to 20 days after a single injection. We attribute this prolonged circulation times to the fact that EVs were isolated from autologous macrophages that may provide their diminished immunogenicity. In contrast, most of the investigations reported focused on the biodistribution of mesenchymal stem cell-derived EVs [30,31], or cancer cells [32,33,34]. Interestingly, the sharp pick of EVs accumulation in the brain was recorded at 72 h, after administration for all studied routes. To confirm these observations and eliminate possible contamination by EVs circulated in the blood stream, BD mice were sacrificed at the end point (480 h), perfused, and main organs were imaged by IVIS. Postmortem organ images reviled highest EVs accumulation in the brain after *i.t.* injection, and in liver and spleen after *i.v.* and *i.p.* injections. Of note, we reported earlier a preferential co-localization of EVs with neurons, microglia, and, partially, with endothelial cells in the inflamed brain [10]. Along with distinct endosomal compartments filled with EVs, a diffuse staining was evident throughout all brain tissues.

Next, to examine the ability of EVs nanocarriers to increase TPP1 accumulation in different organs, CLN2 KO mice were injected with EV-TPP1 via different routes, sacrificed at the peak of EVs accumulation (72 h), perfused, and slides of main organs were examined by confocal microscopy. In these settings, we traced TPP1 fluorescently labeled through covalent conjugation with Alexa 555, as well as EVs nanocarriers labeled through the incorporation of hydrophobic DID dye into EVs membranes. The accumulation brain levels of TPP1 incorporated into EVs was compared to the enzyme injected without EVs. The data clearly demonstrated that EVs facilitated TPP1 transport to the brain, especially after *i.t.* injection. In accordance with IVIS data, *i.v.* and *i.p.* injections of EV-TPP1 resulted in significant TPP1 increases in peripheral organs, especially liver and spleen. Of note, along with the brain delivery, TPP1 transport to the peripheral tissues is also crucial for the attenuation of tissue damage by lysosomal storage material. Therefore, *i.v.* and *i.p.* administration routes should have a great potential for ERT as well. Furthermore, spinal cord pathology contributes significantly to the clinical disease and can be an effective therapeutic target. In this regard, *i.t.* injection that resulted in the increased TPP1 levels in the spinal cord is of great importance.

Consequently, we compared the therapeutic efficacy of EV-TPP1 treatments using four administration routes (*i.v, i.p., i.t.*, and *i.n.*) The main parameters, including accumulation of storage material in the brain tissues, neuroinflammation and neuronal survival were examined in BD mice. All treatment routes provided significant therapeutic effects resulting in decreased astrocytosis and increased neuronal survival, especially in the case of *i.t*. and *i.n.* injections. In addition, the superior reduction of lysosomal storage material in the brain was observed after *i.t*. injections of EV-TPP1. 

Based on the obtained data, we selected a combination of two administration routes, *i.t.* and *i.p.* injections, to achieve efficient TPP1 delivery to CNS and main peripheral organs, respectively, and examined their therapeutic effects on the lifespan of CLN2 KO mice. Significant increases in BD mice survival were recorded when this dual treatment regimen was applied. Furthermore, increases in lifespan were also detected when only *i.p.* treatments with EV-TPP1 were applied, although at a lesser extent than the combination therapy. We speculated that besides the dose of therapeutic enzyme delivered to the organ, the route of administration could also affect the mechanism, and eventually the therapeutic efficacy of the EV-TPP1. In particular, it was reported that autophagy, a complex pathway regulated by numerous signaling events, is involved in the recycling of macromolecules, and therefore, can be perturbed in LSDs [35]. Thus, EV-TPP1 treatments may affect induction of this important process, and therefore, decrease neuronal cell death. This warrants the further in-depth mechanistic investigations of EV-based ERT effects for BD therapy.

In conclusion, ERP is a straightforward and efficient therapeutic approach for lysosomal storage disorders. BioMarin Pharmaceutical Inc. has developed enzymatically active TPP1, Brineura, that is available now for patient’s treatment through intracerebroventricular (*i.c.v*.) injections. The ultimate goal of our investigations is to develop a next-generation Brineura product using natural nanocarriers, EVs. From a clinical perspective, the benefits could include reduced dosage and administration through different and less invasive routes, or their combination with minimal efficacy loss compared to *i.c.v*., or/and potentially improved efficacy at a given dosage vs. the same quantity of free drug. Furthermore, the incorporation of TPP1 into EVs nanocarriers should diminish TPP1 exposure to the immune system, and as a result, improve its therapeutic efficacy and decrease offsite toxicity. This is especially vital in the case of multiple life-long treatments of BD patients. We suggest that different regimens using EVs-based formulations of TPP1 should be considered according to the age of patients, the disease progression, and pathological manifestations. 

## Figures and Tables

**Figure 1 cells-09-01273-f001:**
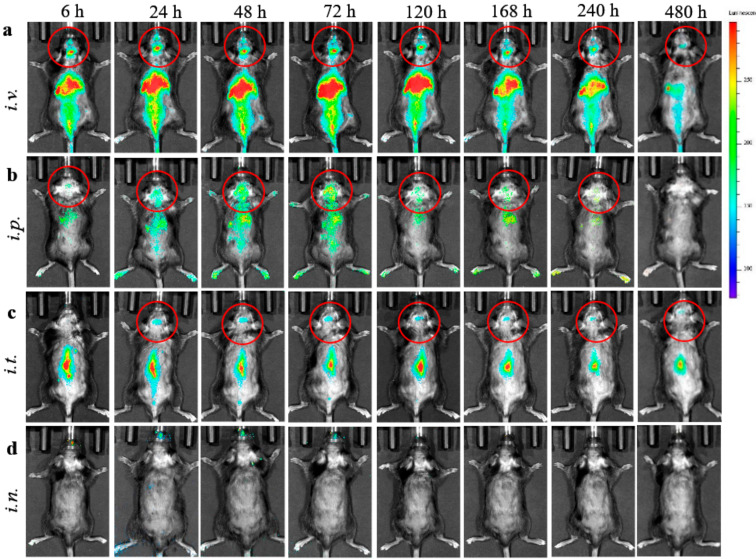
Biodistribution of 1,1’-Dioctadecyl-3,3,3’,3’-tetramethylindo-carbocyanine iodide (DIR)- extracellular vesicles (EVs) in CLN2 KO mice by IVIS. DIR-labeled EVs were administered in BD mice (1 month old, *N* = 4), though: (**a**) *i.v.* (6 × 10^11^ particles/200 µL), (**b**) *i.p.* (6 × 10^11^ particles/200 µL), (**c**) *i.t.* (1.5 × 10^11^ particles/50 µL), or (**d**) *i.n.* (6 × 10^10^ particles/20 µL) routes, and imaged up to 480 h. Prone representative images show prolonged DIR signal accumulation in the brain for *i.v.*, *i.p.*, and *i.t.* administration routs, especially at 24–120 h. Little, if any, DIR signal was observed in live animals after *i.n.* administration (**d**). Accumulation of labeled EVs was also observed in the main peripheral organs for *i.v.* and *i.p.* injections.

**Figure 2 cells-09-01273-f002:**
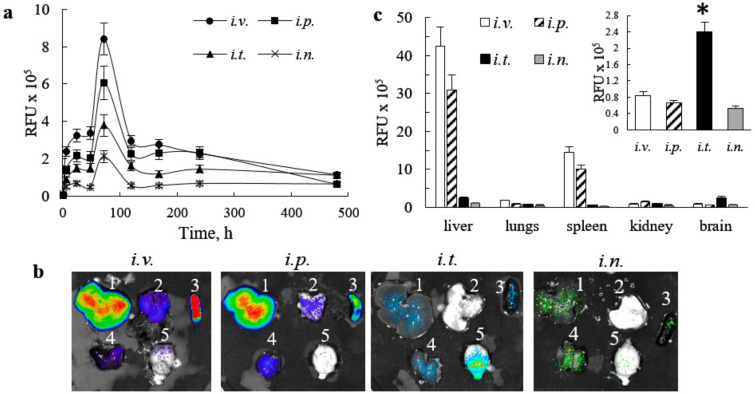
Quantification of DIR-EVs distribution (**a**) in the brain in live animals at various points, and (**b,c**) necroscopy at 480 h in the whole organs in CLN2 Batten disease (BD) mice by IVIS. The groups of four animals BD mice were injected with DIR-EVs, as described in Figure 1. (**a**) The greatest DIR-EVs signals were detected in BD mouse brain after *i.v*., and *i.p*. injections at 72 h. (**b**) At the endpoint (480 h), mice were sacrificed, perfused, and main organs (i.e., liver (1), lungs (2), spleen (3), kidney (4), and brain (5)) were imaged by IVIS, and (c) quantification of DIR-EVs signals in the organs of BD mice was assessed by IVIS Aura software. The highest DIR-EVs signals were recorded in liver, and spleen in mice after *i.v.* and *i.p.* injections. The amount of EVs carriers in the brain of BD injected through *i.t.* route was significantly greater compared to all other injections (insert). Values are the means ± SEM, *N* = 4, * *p* < 0.05.

**Figure 3 cells-09-01273-f003:**
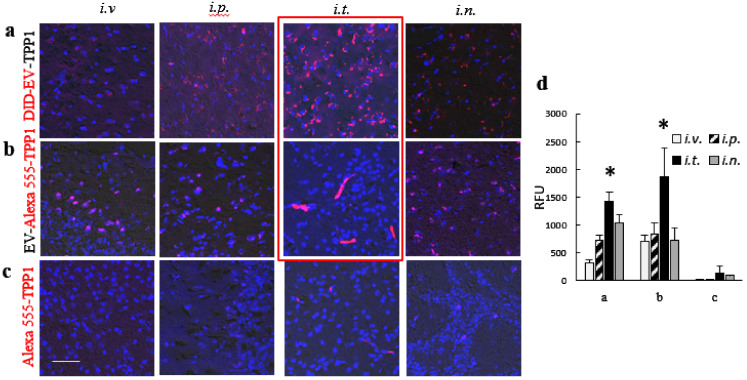
Brain distribution of fluorescently labeled (**a**,**b**) EV-TPP1 formulations and (**c**) TPP1 alone in BD mice. CLN2 KO mice (1 month old, *N* = 4) were administered with: (**a**) DID-EVs (red) loaded with non-labeled TPP1, or (**b**) non-labeled EVs loaded with fluorescently labeled Alexa 555-TPP1 (red), or (**c**) Alexa 555-TPP1 alone without EVs (red) through *i.v.* (2 × 10^10^ particles/200 µL), or *i.p.* (2 × 10^10^ particles/200 µL), or *i.t.* (5 × 10^9^ particles/50 µL), or *i.n.* (2 × 10^9^ particles/20 µL) routs. 72 h later, mice were sacrificed, perfused, brain slides were processed, and examined by confocal microscopy. Nuclei were stained with DAPI (blue). Obtained confocal images (**a**–**c**), and quantification of fluorescence signals (**d**) indicate the significant accumulation of EVs nanocarriers (**a,d**), as well as TPP1 in EVs (**b,d**) in the mouse brain for all administration routes, especially after *i.t.* injection. Little, if any, TPP1 fluorescence was found in the brain when TPP1 was administered alone without EVs nanocarriers (**c, d**). The bar: 50 µm. Values are the means ± SEM, * *p* < 0.05.

**Figure 4 cells-09-01273-f004:**
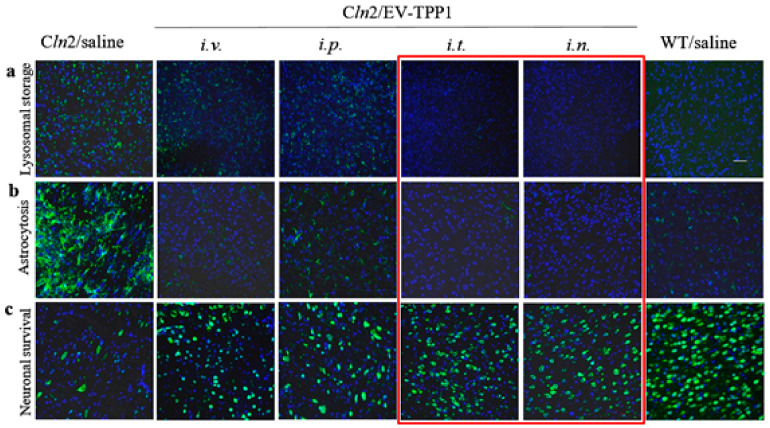
Therapeutic effects of EV-TPP1 formulation administered via different routes in BD mouse model. CLN2 KO mice (1 month old, *N* = 6) were treated with EV-TPP1 through different routes two times a week for three weeks through *i.v*., or *i.p.* routes (4.5 × 10^10^ particles/ 150 µL/mouse; 15 mg/kg TPP1), or *i.t.* route (1.5 × 10^10^ particles/50 µL/mouse; 5 mg/kg TPP1), or *i.n.* route (0.6 × 10^10^ particles/20 µL/mouse; 2 mg/kg TPP1). BD mice and WT littermates injected with saline were used in control groups. Then, mice were sacrificed, perfused, brains were harvested, and slide were stained with: (**a**) Ab to lysosomal storage bodies (subunit C of mitochondrial ATP synthase); or (**b**) Ab to activated astrocytes (GFAP), or (**c**) Ab to neurons (NeuN), and secondary antibody goat anti-rabbit IgG H+L Alexa Flour 488 (green). Slides were washed and mounted with DAPI staining (blue). Treatments with EV-TPP1 via all examined routes resulted in reduced accumulation of lysosomal storage material in the brain (**a**) and neuroinflammation (**b**), as well as decreased neurodegeneration (**c**). Quantification of fluorescent levels (see Appendix A) indicates that *i.t.* and *i.n.* routes provided the most efficient therapeutic effects. The bar: 50 µm.

**Figure 5 cells-09-01273-f005:**
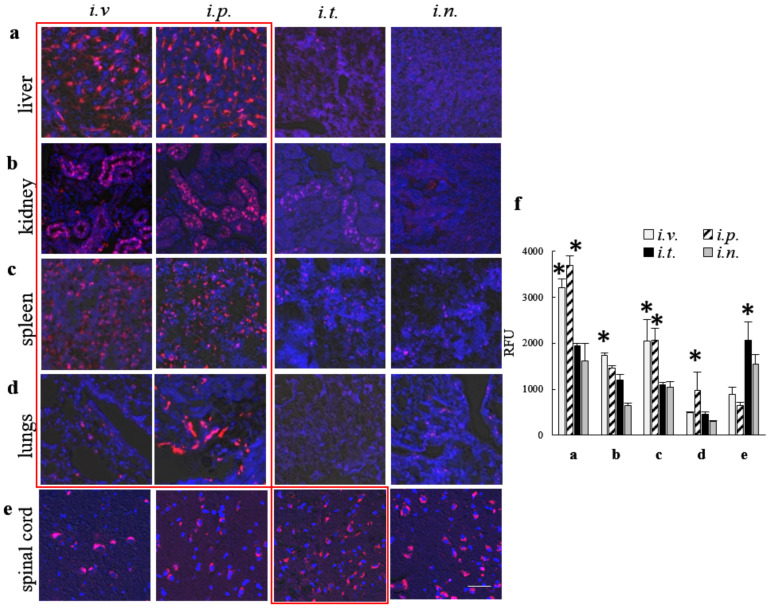
Distribution of fluorescently labeled EV-TPP1 formulations in main organs upon various routes of administration in BD mice. CLN2 KO mice (1 month old, *N* = 4) were injected with non-labeled EVs loaded with Alexa 555-TPP1 (red), through: *i.v.* (2 × 10^10^ particles/200 µL), or *i.p.* (2 × 10^10^ particles/200 µL), or *i.t*^.^ (5 × 10^9^ particles/50 µL), or *i.n.* (2 × 10^9^ particles/20 µL) routs. Then, 72 h later, mice were sacrificed, perfused, main organs, namely: (**a**) liver, (**b**) kidney, (**c**) spleen, (**d**) lungs, and (**e**) spinal cord were processed as described on Figure 3, and examined by confocal microscopy. Nuclei were stained with DAPI (blue). Confocal images (**a**–**e**), and quantification of Alexa 555 fluorescent signals (**f**) indicated significant accumulation of Alexa 555-TPP1 incorporated into EVs in all peripheral organs, especially after *i.v.* and *i.p.* injections. The high Alexa 555-TPP1 signals were recorded in spinal cord after *i.t.* injection (**e**,**f**). The bar: 50 µm. Values are the means ± SEM, * *p* < 0.05.

**Figure 6 cells-09-01273-f006:**
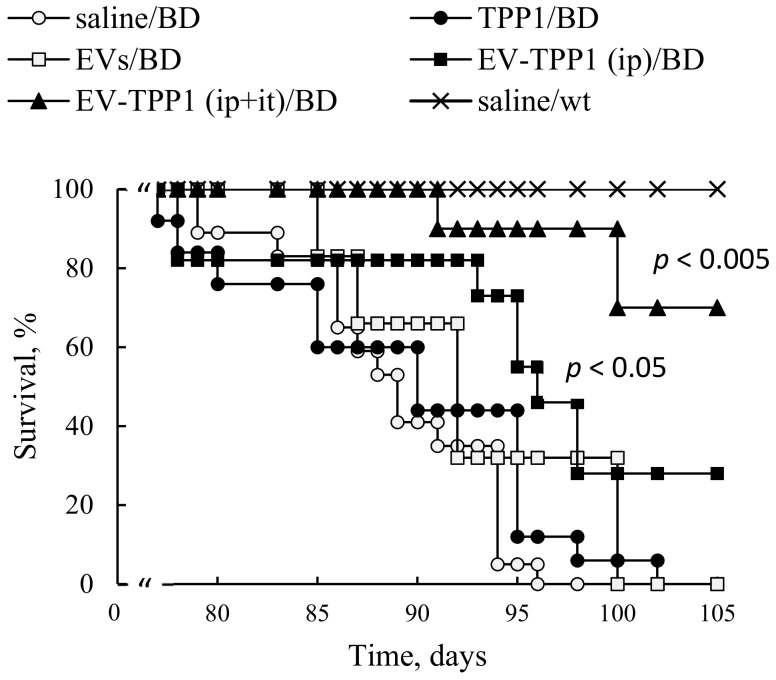
EVs-TPP1 treatments increased lifespan in BD mice. CLN2 KO mice (1 week-old) were treated with *i.p.* injections of: EV-TPP1 (filled squares, 4.5 × 10^10^ particles/150 µL/mouse, 15 mg/kg TPP1) twice a week for six weeks; or EVs alone (empty squares, the same dose of particles); or TPP1 alone (filled circles, the same dose of TPP1); or saline (empty circles). Another group of CLN2 KO mice received the same dose of EV-TPP1 through *i.p.* route, and *i.t.* injections of EV-TPP1 at second week (filled triangles, 1.5 × 10^10^ particles/50 µL/mouse; 5 mg/kg TPP1) once a week, for five weeks. Control WT animals were *i.p.* injected with saline (crosses). A lifespan was recoded over three months. A significant increase in lifespan was demonstrated for CLN2 KO mice treated with EV-TPP1 through *i.p.,* and especially through a combination of *i.p.* and *i.t.* injections, compared to those treated with saline, or TPP1 alone, or sham EVs (*N* = 6).

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
