# Peer review of "Extracellular Vesicles as Drug Carriers for Enzyme Replacement Therapy to Treat CLN2 Batten Disease: Optimization of Drug Administration Routes"

_cells, 2020, doi:10.3390/cells9051273_

Round 1

Reviewer 1 Report

In this work Haney et al,  used extracellular vesicles as drug carriers to deliver TPP1 to brain and other tissues of batten disease mouse model. They demonstrate that this approach effectively lead to the delivery of thee recombinant enzyme to target tissues providing potential therapeutic benefits.

This work represent a partial extension of a previously published work by the same group (see Haney et al. Adv. Healthcare Mater.2019, 8, 1801271). In the current work they focus on brain and compared different therapeutic administration routes.  However in most cases the data presented are still very preliminary and not very convincing and should be significantly improved. In particular the following aspects needs to be carefully addressed:

  • In vivo neuronal pathology should be evaluated, in particular lysosomal morphology and storage should be compared in neurons from treated vs untreated animals.
  • Distribution of the enzyme: In figure 3 randomly chosen area were chosen. A more systematic study is needed and distribution of the enzyme should be carefully evaluated comparing  different and defined brain regions.
  • Subcellular distribution of the enzyme: the authors should evaluate distribution of the enzyme in neurons, astrocyte and glia, using appropriate immunofluorescent markers.
  • Figure 5 and 6 show enzyme delivery, but again no data relative to the rescue of cellular and tissue pathology were presented.

Reviewer 2 Report

In a recent paper, this group optimized a method for production macrophage-derived extracellular vesicles, loaded with recombinant TPP1 protein (EV-TPP1), and demonstrated therapeutic effect after ip injection to TPP1-/- BD mice. This current paper extends investigation to compare various administration routes for EV distribution and therapeutic effect of EV-TPP1.

This study provides some promising preliminary data indicating that the EV-TPP1 has therapeutic efficacy for CLN2 BD, perhaps an improvement over direct ERT. Their findings support further exploration EV as vehicles for enzyme replacement therapy.

The IVIS data is very nice, showing clearly that EV signal is retained for quite a while in different tissues dependent on injection route.

Major concerns relate to questionable conclusions drawn from poorly presented histology data for A555-TPP1 detection in tissue sections and staining for SCMAS, GFAP, and NeuN, and the RFU quantitation of this data. These concerns are detailed as follows.

Figure 3 – where in the brain are these images from?  There has to be consistency for fair comparison.  2b it and 2c regions look quite different than others. RFU quantification from images is determined from how many images, from how many brain regions?  Doe the N refer to number of mice?

Figure 4 - where in brain are the images from? Pathology in the CLN2 KO mouse brain is not uniform. The equivalent regions should be imaged across all “routes” for each test (a, b, and c) for fair comparison.  Also, autofluorescence of storage material is a pathological feature.  Since the authors are using immuno-fluorescence, fair comparison necessitates capturing images from the exact same brain region, as well as controlling laser exposure and image capture times.  DAB staining would be preferred staining method to avoid this problem.

The data for preservation of neurons is not convincing. Usually stereology is done to quantify neuron loss.  Staining for NeuN in a few sections from perhaps non-equivalent regions does not convince me that neurons are preserved in treatment groups. What region in the CLN2 ko brain would show significant neuronal loss over this 3-week period?

Figure 6 -Where are these images from? Eye images should show all layers of the retina, and should indicate cell types that pick up EV.   Do ganglion cells take up EV?  Where in brain are these images from? How are EV reaching brain/neurons? For i.o., did you examine brain areas innervated by optic tract axons?  Correction of retinal-related pathology and vision could be a separate study with i.o. route. 

This study would be stronger if tissues were collected at a late time-point from mice treated as in figure 8, to assess TPP1 presence/activity and pathology (scmas, GFAP, etc., should see substantial prevention of brain pathology in several regions of brain, and peripheral tissues).

Minor concerns and comments are listed below.

Line 3. “…Treat Batten Disease…” should be more specific:  “…Treat CLN2 Batten Disease…”

Line 21.  “leaded” should be “led”

Lines 34-40. Batten disease can be caused by mutations in genes other than TPP1.  Mutations in PPT1 (CLN1), TPP1 (CLN2), and CLN3 genes give rise to classical forms of infantile, late infantile and juvenile forms of BD (INCL, LINCL, JNCL), respectively. The authors should consult a recent BD review for current nomenclature.

Lines 47-48. long biweekly infusions of TPP1 have a low patient adherence – can the authors cite a reference for this?

Line 82.  Is the recombinant TPP1 full-length? (could provide uniprot ID).

Line 108. Could the authors provide more detail on the CLN2-/- mouse model: the source of the mice, nature of the mutation, and genetic background.

Line 116.  NTA first use, what does it mean?

Lines 128-129.  What is the model of the sonicator and intensity setting of sonication? The amount of Alexa 555-TPP1 in the EV suspension prior to sonication is not clear; 5ul of a 1ug/ml stock added?

Lines 147-149.  Considerably more detail should be provided for injection procedures, particularly the i.t. and i.o. injections.  The reader should be able to replicate procedures.  We don’t even know if i.t. is cervical or lumbar area.

Line 159. How were tissues processed and sectioned, and how thick are the tissue sections?

Lines 202-203.  “Of note, sonication loading procedure did not decrease TPP1 activity.”  Can you show data? – activity level of TPP1 in the EV compared to activity level of non-EV incorporated TPP1 (normalized to TPP1 protein level as determined by western blot)?

Lines 217-218. “About 54% TPP incorporated into EVs were co-localized with lysosomes after 4h-incubation.” Endogenous Lamp1 and Lamp2 are known to be present in EV.  If these EV incorporate Lamp1, then co-localization to Lamp1 does not demonstrate delivery to lysosomes, and the authors need to take a different approach to verify this.

Lines 381-383. Sentence doesn’t make sense.

Line 398. do you mean "to examine the accumulation of TPP1 in tissues after EV delivery..."?

Line 399 Pick = peak?

Lines 414-415. “Intriguingly, our data implicate that i.o. injection, and even eye drops increased TPP1 accumulation not only in the retina, but also in the brain and spinal cord”.  Agreed that this is intriguing.  Can you discuss how EV might traffic to give these results?

Lines 417-420 These conclusions are questionable.

I am curious as to whether EV-TPP1 application via any of these tested routes leads to the generation of antibodies to TPP1? Age of mice, route of EV injection, as well as cell-source of EV might contribute the generation of an antibody response (or alternatively promote tolerance).  Would this be relevant factor in comparison to direct (non EV) ERT?

Can the authors provide Discussion comparing their treatment to standard ERT, with similar regimen of ip plus it routes of TPP1 injection?

Reviewer 3 Report

The work by Haney et al. has the ambitious aim to address the issue of targeting drugs to the brain in the context of a lysosomal disease caused by deficiency of wild-type TPP1 enzyme. The authors build on their previous research for what concerns the use of extracellular vesicles (EV) as drug carriers, the cell source of EV, and the method for drug loading. The manuscript is overall well-written and results clearly reported. Yet, a major technical issue jeopardize the majority of results. It is now well-known that the use of lyophilic dyes such as DID and DIR (or the more used PKH26) may result in artifacts, such as fluorescent dye micelle or fluorescently labeled protein aggregates. These artifacts are indistinguishable from labeled EV under microscopy/bioimaging analysis and may persist even longer than EV in the tissues. They are also co-isolated with EV by ultracentrifugation. Thus, all biodistribution data herein presented, and partially the brain accumulation data (where another labeling method was applied), could be biased by these artifacts. The brain accumulation data are somewhat validated by the other labeling method applied (AF555-TPP1), but only representative fields of brain tissue were analysed and no data on whole brain or other organs are presented. This issue is of paramount importance in a work focused on biodistribution and brain targeting.

Major concerns

  • The authors need to address the issue of artifacts arising from the use of lipophilic dyes, adding controls for 1) pure dye accumulation and biodistribution in the animal, 2) biodistribution of residual dye coming from the same preparation protocol used for EV, mimicking all passages from supernatant harvest to concentration of EV and administration to animals. An alternative may be to validate biodistribution results using another labeling methodology, or using further purifying protocols after labeling with lipophilic dyes that guarantee removal of artifacts.
  • Authors must follow MISEV2018 guidelines for minimal requirements to define EV (Thery, 2018, JEV). In particular, please provide the following for the EV used in this work: TEM images or similar; western analysis to address purity of EV preparations from material of other organelles (e.g.: golgi, mitochondria, ER). Alternatively, provide references to previous works by the authors where the same EV were characterized as requested.

Minor concerns

Introduction

The introduction would benefit from a bit more background on EV biology (types of EV, biogenesis, rationale of EV as organ targeting carriers)

Materials and Methods

  • SEC methodology should be illustrated
  • Rationale of LAMP1 labeling with primary antibody before permeabilization should be clarified
  • rational for use of DIR for IVIS and DID for confocal microscopy should be disclosed
  • purity of EV preparations as illustrated by Webber and Clayton paper “How pure are your vesicles?” (Webber, 2013, JEV) or with similar methodology should be calculated and added to the manuscript
  • integrity of EV after sonication should be clearly demonstrated and discussed (TEM would be the best option)
  • EV isolation method should be detailed as per MISEV2018 indications
  • AF555 labeling method should be described

Results

  • Line 208: endocytic pathway is one of the possible way EV are uptaken by target cells, the authors should state correctly
  • Line 209-210: literature 15-17 is not related to the statement of the authors. The authors should correct and check that proper literature is referenced throughout the manuscript
  • Details of how quantifications in fig.3d, 5f and 6d were performed should be disclosed (number of fields, from how many animals, and independent experiments)

Discussion

  • relevance of dose-response approach in biodistribution and brain accumulation studies should be discussed. Have the authors ever done that? Will be matter of future investigations?
  • Timing of clearance of EV from organs should be discussed compared to the literature

Round 2

Reviewer 1 Report

The authors have responded to all my comments. Given that a main message of the paper is the delivery of enzyme to the brain, I think that a more careful analysis of enzyme distribution and activity in the different brain regions and cellular populations would have greatly improved the significance of the paper over the existing literature.

Author Response

Thank you very much for all your valuable comments. We will carry out further analysis of the enzyme distribution and activity in the different brain regions and report the results in the following manuscript.

Reviewer 2 Report

The authors addressed most concerns with modifications throughout to substantially improve and strengthen the manuscript.  However, the results presented in Fig.3 and Fig 4 (plus Suppl Table) were not improved and I remain skeptical of this data. It is imperative that images are captured from identical regions, to fairly compare effectiveness of the various EV administration routes for brain targeting and therapeutic effects.  In the normal brain, densities and types of neurons vary dramatically across regions, and in Batten Disease, susceptibility to pathology is not uniform. The authors added to Methods “All images were taken from the exact same brain region (cerebral cortex) with the controlled laser exposure and image capture time.”  Two points about this.  First, some images do not look like cerebral cortex, and second, “cerebral cortex” is not specific enough – images should be captured from the same region of cerebral cortex (example, motor cortex).  The authors added a supplemental figure to include more brain regions for i.t. EV distribution - frontal lobe, cerebral cortex, thalamus, hypothalamus, temporal lobe and “ponds” (should be pons).  Frontal and temporal lobes are regions of the “cerebral cortex”. In looking at these figures, it is apparent that there are duplicate images. Suppl Fig.A5 “hypothalamus” is identical to Fig. 3b-i.t. “cerebral cortex”, and Fig. 3a-i.t. is identical to Fig. 3a-i.n.

Based on fluorescence intensity measures from stained “cerebral cortex” images after 3 wk EV-TPP1 treatment of young mice, the authors claim therapeutic effects: reduction in gliosis and storage, and neuron survival. If the authors were not careful to exactly match brain regions for image capture, then measurements are faulty and conclusions cannot be drawn.  I am particularly skeptical of the neuron loss measurements. Is there literature to support such dramatic cortical neuron loss from 30 days to 51 days old in this model? Again, neuron loss would require stringent matching of brain regions – and would be better determined by neuron counts (rather than mean fluorescence signal).

Reviewer 3 Report

All concerns were addressed or appropriately discussed by the authors.

Author Response

Thank you once again for the thorough and reasoned review of our manuscript. The changes now made clearly serve to improve the impact of the article.